# MFmap: A semi-supervised generative model matching cell lines to tumours and cancer subtypes

Xiaoxiao Zhang[1,2], Maik Kschischo[1]*

**1** Department of Mathematics and Technology, RheinAhrCampus, University of Applied Sciences Koblenz, Remagen, Germany, **2** Department of Informatics, Technical University of Munich, Munich, Germany

* kschischo@rheinahrcampus.de

**Data Availability Statement:** All data used in this study are publicly available online. Detailed references to access the data can be found in the main text and Supplemental Information. We have also added a cloud folder at where all preprocessed

## Abstract

Translating *in vitro* results from experiments with cancer cell lines to clinical applications requires the selection of appropriate cell line models. Here we present MFmap (model fidelity map), a machine learning model to simultaneously predict the cancer subtype of a cell line and its similarity to an individual tumour sample. The MFmap is a semi-supervised generative model, which compresses high dimensional gene expression, copy number variation and mutation data into cancer subtype informed low dimensional latent representations. The accuracy (test set $F_1$ score >90%) of the MFmap subtype prediction is validated in ten different cancer datasets. We use breast cancer and glioblastoma cohorts as examples to show how subtype specific drug sensitivity can be translated to individual tumour samples. The low dimensional latent representations extracted by MFmap explain known and novel subtype specific features and enable the analysis of cell-state transformations between different subtypes. From a methodological perspective, we report that MFmap is a semi-supervised method which simultaneously achieves good generative and predictive performance and thus opens opportunities in other areas of computational biology.

## Introduction

Tumour-derived cell lines are important model systems for developing new anti-cancer treatments and for understanding cancer biology [1–3]. They are comparably cost efficient, easy to handle under laboratory conditions and do not inflict ethical issues arising in research involving human or animal subjects. Yet, promising cell line experiments are rarely translated to clinical applications. In some cases, there are remarkable differences between cell lines and the primary tumours they were derived from [2–4]. This is also the reason why the assignment of clinically informative tumour subtypes to cell line models [3–5] is not a straightforward task.

To narrow the gap between preclinical findings and tumour treatment, it is necessary to select appropriate cell line models for a given tumour sample or a given cancer subtype. Several attempts to evaluate similarities and differences between cell lines and bulk tumours have

data are available: https://cloud.hs-koblenz.de/s/ytFKkzcK78AekL4.

**Funding:** This work was supported by the FOR2800 research unit funded by the Deutsche Forschungsgemeinschaft (DFG project number 395736209).The funders had no role in study design, data collection and analysis, decision to publish, or preparation of the manuscript.

**Competing interests:** The authors have declared that no competing interests exist.

focused on associations between corresponding data modalities including mutation, copy number, gene expression and methylation [6–12]. An important data resource comes from collaborative projects like NCI-60 [13] and the Cancer Cell Line Encyclopaedia (CCLE) [5, 14], who have generated large-scale pharmacogenomics data from patient-derived cell lines across organs. Other efforts like Sanger Genomics of Drug Sensitivity in Cancer (GDSC) [15], Connectivity Map (CMAP) [16], the Cancer Therapeutics Response Portal (CTRP v1 and CTRP v2) [17, 18] further expanded the datasets. On the other hand, The Cancer Genome Atlas (TCGA) [19] and the International Cancer Genome Consortium (ICGC) [20] systematically characterised molecular profiles of thousands of tumours. These complementary data resources are valuable for understanding the complexity of cancer biology and connecting *in vitro* pharmacogenomic profiles to patient molecular characteristics, potentially informing anti-cancer treatment strategies.

Integrative analyses considering multiple data types of both cell lines and bulk tumours are still challenging and new analysis concepts tailored towards specific questions are an ongoing research topic. For instance, Cellector [21] preselects the most frequent genomic alterations and defines cancer subtypes based on a sequence of these alterations. Although such a preselection of genomic alterations integrates prior knowledge about cancer mutational patterns, it neglects complementary information contained in other data types. Furthermore, Cellector relies on a binary matrix of genomic alterations. This matrix is very sparse, since samples harbouring the same alterations are very rare. Therefore, the statistical power to detect appropriate cell lines for tumours might be limited.

A recent study [22] highlighted that independent classifiers based on different data types to predict cell line identity often yield inconsistent results. For example, predictions based on the mutation spectrum and oncogenic mutations can be contradictory, although both features are derived from mutation data. Complementary information from different data sources is integrated by the MAGNETIC-framework [23] into gene modules. Gene set enrichment analysis (GSEA) is then used to interpret these modules as pathways. MAGNETIC is indeed a powerful technique for integrating multiple molecular datasets and prior knowledge, but it does not conclude to what extent a cell line is suitable as a tumour model. The maui framework assigns cancer subtype labels to cell lines by extracting relevant features from multiple data types using a variational autoencoder (VAE) [24]. However, most of the maui embedded features are weakly associated with subtype labels and are therefore difficult to interpret.

Here, we propose MFmap, a new semi-supervised VAE architecture and objective function which combines good classification accuracy with good generative performance. We exploit these properties to derive subtype informed low dimensional representations for both cell lines and bulk tumours from high dimensional multi-omics data including gene expression, mutation and copy number variation. The latent representations can then be used to assess the similarity between a cell line and a tumour. We provide cell line by tumour dissimilarity matrices for CCLE and TCGA for the ten different cancer types listed in Table 1. In addition, MFmap predicts cancer subtype labels for cell lines. We demonstrate, how these predicted cancer subtypes can be used to transfer information from cell-line-based drug sensitivity screens to patient cohorts. We also show, that the latent representations learnt by MFmap are biologically interpretable. Finally, we illustrate how the generative nature of the MFmap model can be exploited for studying subtype transformations during cancer progression. At http://h2926513.stratoserver.net:3838/MFmap_shiny/ we provide a resource enabling researchers to select the most relevant cell line for a cancer patient.

**Table 1. The sample size of TCGA and CCLE data used for training and testing MFmap.**

| TCGA code | study name | number of subtypes | TCGA sample size | CCLE sample size |
|---|---|---|---|---|
| BRCA | Breast invasive carcinoma | 4 | 484 | 51 |
| COADREAD | Colon adenocarcinoma | 4 | 414 | 54 |
| ESCA | Esophageal carcinoma | 2 | 169 | 27 |
| HNSC | Head and neck squamous cell carcinoma | 4 | 278 | 29 |
| LUAD | Lung adenocarcinoma | 3 | 227 | 70 |
| LUSC | Lung squamous cell carcinoma | 4 | 178 | 22 |
| PAAD | Pancreatic adenocarcinoma | 2 | 149 | 40 |
| SKCM | Skin cutaneous melanoma | 3 | 260 | 49 |
| UCEC | Uterine corpus endometrial carcinoma | 3 | 234 | 28 |
| GBMLGG | Glioblastoma multiforme and lower grade glioma | 7 | 621 | 55 |

## Materials and methods

### Matching cell lines and tumours as a semi-supervised learning problem

MFmap is a semi-supervised deep neural network which integrates gene expression, copy number variation (CNV) and somatic mutation data with subtype classification. Each tumour sample $t$ consists of a pair of $(x_t, y_t)$, where $x_t \in \mathbb{R}^D$ denotes the high dimensional molecular features and $y_t \in \{1, \ldots, h\}$ is the cancer subtype label. For a cell line $c$, the cancer subtype is unknown and only the molecular features $x_c$ are available. The index $c$ or $t$ will be suppressed, whenever we refer to a single observation. The MFmap neural network is trained in a semi-supervised manner using both cell line data $\mathcal{D}_{cl}^{train} = \{x_c\}_{c=1}^{C_{train}}$ and tumour data $\mathcal{D}_{tu}^{train} = \{(x_t, y_t)\}_{t=1}^{T_{train}}$. Here, we used cell line data from CCLE and tumour data from TCGA.

One aim of MFmap is to use semi-supervised classification to infer the cancer subtype $y_c$ of a cell line $c$. A second aim is to assess the similarity between a cell line and a tumour. Instead of comparing the high dimensional molecular features $x_t$ and $x_c$ directly, we first encode them into low dimensional latent representations $z$ (see next section for details). Then, the similarity of a tumour sample $t$ and a cell line $c$ is measured as the cosine coefficient between the corresponding latent representation vectors $z_t$ and $z_c$. We will also show that these latent representations $z$ carry interpretable biological information.

The molecular data $x = (x_{DNA}, x_{RNA})$ consist of gene expression profiles $x_{RNA}$ and network smoothed mutation and CNV profiles $x_{DNA}$. We will refer to these two parts as RNA and DNA view, respectively. The DNA view is obtained from the original binary mutation and CNV matrices (Fig 1(A)), which indicate the occurrence of a mutation or CNV event targeting a gene in a given tumour sample or cell line. These very sparse matrices are first projected onto an annotated cancer network [25]. By using a network diffusion algorithm [26], a mutation or CNV signal hitting a single gene is propagated to neighbouring nodes in the network, thereby enriching the mutation or CNV data by cancer network information. All molecular features were translated and scaled to the interval between zero and one.

### Specification of MFmap as a semi-supervised generative model

The MFmap neural network (Fig 1(B)) is a new variant of a semi-supervised VAE [27]. The observable data are considered to be drawn from the probability distributions $p(x, y)$ for tumour samples and $p(x)$ for cell lines. These distributions are modelled as marginals over the

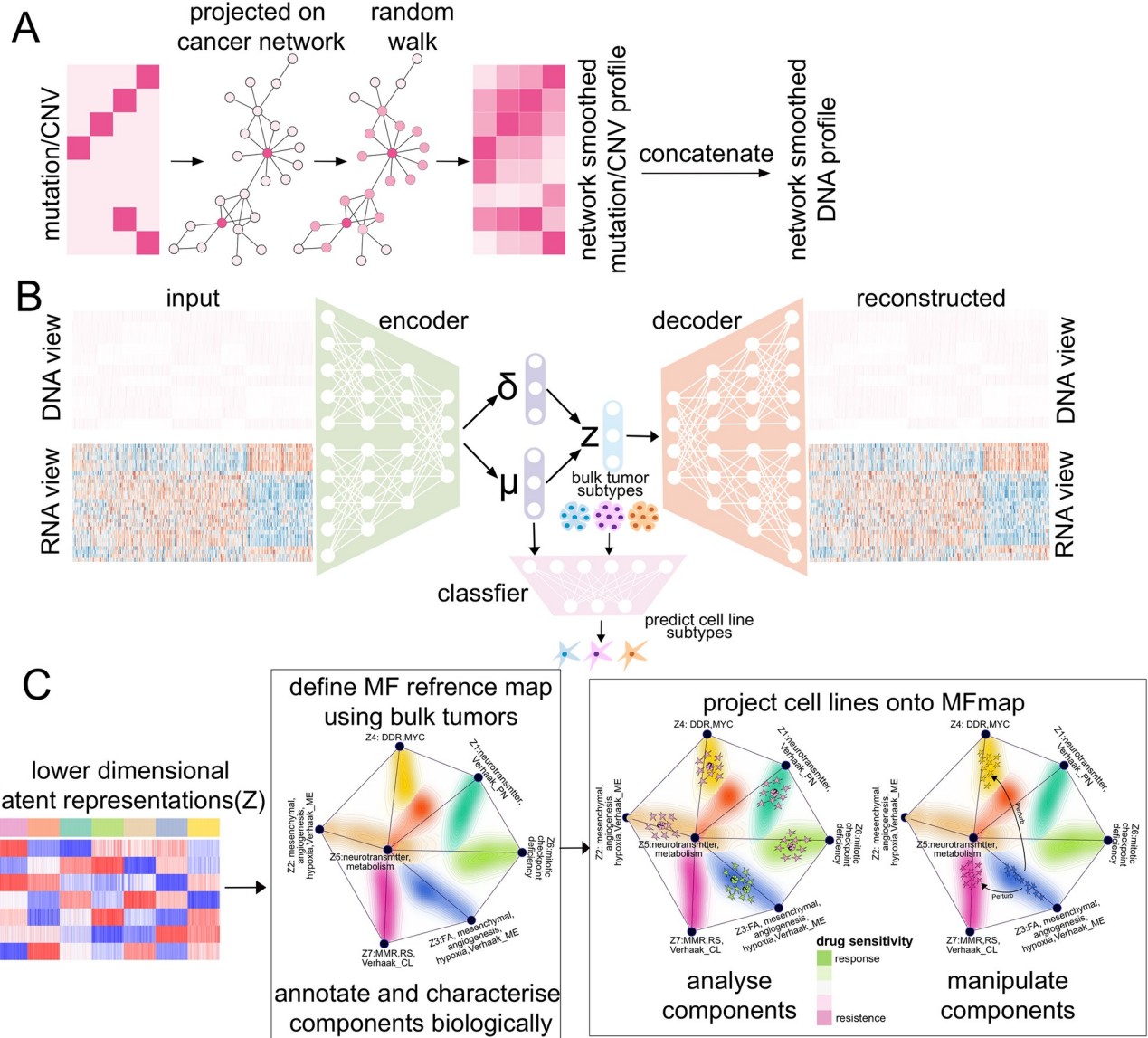

**Fig 1. Overview of MFmap.** (A) In a preprocessing step, mutation and CNV profiles are transformed to network smoothed DNA profiles. The original mutation and CNV data are represented as a binary matrix indicating the presence/absence of a DNA alteration in a given tumour sample or cell line. This sparse matrix is projected onto a cancer reference network (CRN) [25] and a network diffusion algorithm propagates this information to network neighbours, resulting in a dense DNA mutation or CNV matrix (DNA features). (B) The smoothed DNA features (DNA view) combined with gene expression data (RNA view) form the input of MFmap. The neural network architecture of MFmap has three components: encoder, decoder and classifier, encoded by different colours. The encoder maps sample features to a distribution $q(z|x)$ for the latent representation $z$ with mean value $\mu(x)$ and covariance $\sigma^2(x)$. The classifier outputs a molecular subtype probability $p(y|z)$ and the decoder models a density $p(x|z)$ for the reconstruction of the DNA and RNA views. During semi-supervised training, the molecular subtypes of tumour samples are used. (C) For visualisation, the latent representations of bulk tumour samples are used to generate a reference map. Cell lines are then projected to the reference map. The colour coding of individual samples or cell lines (dots) indicates the tumour subtype or the predicted subtype, respectively. The density of the tumour samples is indicated by background contour lines coloured according to the subtypes.

latent variable $z = (z_1, \ldots, z_d)^T \in \mathbb{R}^h$, such that

$$p(\boldsymbol{x}, y) = \int p(\boldsymbol{x}, y, \boldsymbol{z}) \, dz, \qquad p(\boldsymbol{x}) = \sum_{y=1}^{h} p(\boldsymbol{x}, y). \qquad (1)$$

To facilitate biological interpretation of the latent representations, we set the dimension $d$ of the latent space equal to the number of cancer subtypes $h$. In other applications of the MFmap model, one could also consider $d$ as a tuneable hyper-parameter.

For the generative model, we assume $\boldsymbol{x}$ and $y$ to be conditionally independent given the latent variable $z$. Accordingly, the joint distribution can be factorised as

$$p(\boldsymbol{x}, y, \boldsymbol{z}) = p(\boldsymbol{x}|\boldsymbol{z}) \, p(y|\boldsymbol{z}) \, p(\boldsymbol{z}). \qquad (2)$$

These distributions are specified as

$$
\begin{align}
p(\boldsymbol{z}) &= \mathcal{N}(\boldsymbol{z}|0, \mathrm{I}) & (3a) \\
p(y|\boldsymbol{z}) &= \mathrm{Cat}(y|\boldsymbol{\pi_\theta}(\boldsymbol{z})) & (3b) \\
p(\boldsymbol{x}|\boldsymbol{z}) &= \boldsymbol{f_\theta}(\boldsymbol{x}|\boldsymbol{z}). & (3c)
\end{align}
$$

Here, $p(\boldsymbol{z})$ is the prior distribution for the latent representation vector. We denote the Gaussian distribution with mean vector $\boldsymbol{\mu}$ and covariance matrix $\Sigma$ by $\mathcal{N}(\cdot|\boldsymbol{\mu}, \Sigma)$. The parameter $\boldsymbol{\pi_\theta}(\boldsymbol{z})$ of the categorial distribution $p(y|\boldsymbol{z})$ depends on the latent representation $\boldsymbol{z}$. For the decoder $p(\boldsymbol{x}|\boldsymbol{z})$ one can chose a suitable distribution $\boldsymbol{f_\theta}$ with parameters depending on the latent representations $\boldsymbol{z}$ [27]. The functions $\boldsymbol{z} \mapsto \pi_\theta(\boldsymbol{z})$ and $\boldsymbol{z} \mapsto \boldsymbol{f_\theta}(\cdot|\boldsymbol{z})$ are represented as neural networks. The parameters of these decoder networks are jointly denoted as $\boldsymbol{\theta}$.

For the mfMAP model we initially used a Gaussian distribution $\boldsymbol{f_\theta}(\boldsymbol{x}|\boldsymbol{z})$ to model the outputs. However, we found that rescaling the molecular features $\boldsymbol{x}$ to the interval [0, 1] and using a Bernoulli distribution for $\boldsymbol{f_\theta}$ improved the semi-supervised classification accuracy (see Results section). Then, each single output of the decoder neural network $\boldsymbol{z} \mapsto \boldsymbol{f_\theta}(\cdot|\boldsymbol{z})$ can be interpreted as the probability, that the corresponding molecular feature is active or not. For instance, for the $i$−th component $(\boldsymbol{x_{RNA}})_i$ of the RNA-view, the corresponding output can be regarded as the probability that the $i$-th gene is expressed.

Posterior inference, i.e. the evaluation of $p(y, \boldsymbol{z}|\boldsymbol{x})$ using Bayes theorem, is often intractable, because the marginal likelihood $p(\boldsymbol{x})$ in Eq (1) requires integrating over $\boldsymbol{z}$. Therefore, a variational distribution $q(y, \boldsymbol{z}|\boldsymbol{x})$ is introduced to approximate the true posterior [24, 27]. We assume that the variational distribution reflects the conditional independence $\boldsymbol{x} \perp y|\boldsymbol{z}$ of the generative model in Eq (2). This implies

$$q(\boldsymbol{x}, y|\boldsymbol{z}) = q(\boldsymbol{x}|\boldsymbol{z}) \, q(y|\boldsymbol{z}). \qquad (4)$$

For consistency we assume that $q(y|\boldsymbol{z})$ in Eq (4) is identical to $p(y|\boldsymbol{z})$ in Eq (3b) and is represented by the same neural network mapping $\boldsymbol{z}$ to the categorial parameter $\boldsymbol{\pi_\theta}(\boldsymbol{z})$. For the variational distribution $q(\boldsymbol{z}|\boldsymbol{x})$ we choose a Gaussian

$$q_\phi(\boldsymbol{z}|\boldsymbol{x}) = \mathcal{N}(\boldsymbol{z}|\boldsymbol{\mu}, \mathrm{diag}(\boldsymbol{\sigma})) \qquad \text{with} \qquad (\boldsymbol{\mu}(\boldsymbol{x}), \log\boldsymbol{\sigma}(\boldsymbol{x})) = \boldsymbol{g_\phi}(\boldsymbol{x}) \qquad (5)$$

with parameters $\boldsymbol{\mu}(\boldsymbol{x})$ and $\boldsymbol{\sigma}(\boldsymbol{x})$. The parameters are represented by the encoder neural network $\boldsymbol{g_\phi}$, which is itself parametrised by $\phi$. The overall architecture of MFmap (Fig 1(B)) is thus formed by three neural networks, the encoder Eq (5), the classifier Eq (3b) and the decoder Eq (3c).

## Training of MFmap using a semi-supervised loss function

Variational inference involves maximising an evidence lower bound (ELBO) to the log-likelihood of the observational data [24, 27]. For a single cell line sample $\boldsymbol{x}_c \in \mathcal{D}_{cl}$ one can derive a lower bound to the log-likelihood

$$\log p(\boldsymbol{x}_c) = \log\left(\sum_y \int p(\boldsymbol{x}_c, y, \boldsymbol{z}) d\boldsymbol{z}\right) \geq \mathfrak{L}(\boldsymbol{x}_c), \tag{6}$$

which is identical to the ELBO of the basic VAE [24] for unsupervised learning

$$\mathfrak{L}(\boldsymbol{x}) = E_{q(\boldsymbol{z}|\boldsymbol{x})}[\log p(\boldsymbol{x}|\boldsymbol{z})] - D_{KL}(q(\boldsymbol{z}|\boldsymbol{x})||p(\boldsymbol{z})), \tag{7}$$

consisting of a reconstruction loss term and a Kullback-Leibler (KL) divergence term. For a single labelled tumour sample $(\boldsymbol{x}_t, y_t) \in \mathcal{D}_{tu}$ we have for the log-likelihood

$$\log p(\boldsymbol{x}_t, y_t) = \log\left(\int p(\boldsymbol{x}_t, y_t, \boldsymbol{z}) d\boldsymbol{z}\right) \geq \mathfrak{L}_{tu}(\boldsymbol{x}_t, y_t), \tag{8}$$

where the ELBO for labelled examples reads

$$\mathfrak{L}_{tu}(\boldsymbol{x}, y) = \mathfrak{L}(\boldsymbol{x}) + E_{q(\boldsymbol{z}|\boldsymbol{x})}[\log p(y|\boldsymbol{z})]. \tag{9}$$

To derive this ELBO (see S1 File), we exploited the conditional independence assumption $\boldsymbol{x} \perp y|\boldsymbol{z}$ for both the generative model (Eq (2)) and the inference model (Eq (4)). The additional term in Eq (9) in comparison to Eq (7) can be interpreted as a classification loss. Given a tumour sample $(\boldsymbol{x}_t, y_t)$, the probability for the cancer subtype label $p(y_t|\boldsymbol{z})$ is a function of $\boldsymbol{z}$, which is inferred from $q(\boldsymbol{z}|\boldsymbol{x}_t)$. This distribution is in turn determined by the molecular feature vector $\boldsymbol{x}_t$.

We found empirically that the semi-supervised classification accuracy during training was relatively poor when using these exact negative ELBOs as loss functions. This is in line with previous findings that achieving both good semi-supervised classification accuracy and good generative performance is often difficult in VAEs [28] or other generative models [29]. Motivated by the work from [30], we added the negative entropy $\mathcal{H}[p(y|\boldsymbol{z})]$ of the distribution $p(y|\boldsymbol{z})$ to the unsupervised ELBO $\mathfrak{L}$ in Eq (7) and to the supervised ELBO $\mathfrak{L}_{tu}$ in Eq (9). In summary, the MFmap loss functions for the unlabelled cell line and the labelled tumour data are respectively given by

$$\begin{aligned}
\mathcal{U}(\boldsymbol{x}) \quad &= -\mathfrak{L}(\boldsymbol{x}) + \mathcal{H}[p(y|\boldsymbol{z})] \\
&= -E_{q(\boldsymbol{z}|\boldsymbol{x})}[p(\boldsymbol{x}|\boldsymbol{z})] + D_{KL}(q(\boldsymbol{z}|\boldsymbol{x})||p(\boldsymbol{z})) + \mathcal{H}[p(y|\boldsymbol{z})] \quad &\text{(10a)} \\
\mathcal{S}(\boldsymbol{x}, y) \quad &= -\mathfrak{L}_{tu}(\boldsymbol{x}) + \mathcal{H}[p(y|\boldsymbol{z})] \\
&= -E_{q(\boldsymbol{z}|\boldsymbol{x})}[p(\boldsymbol{x}|\boldsymbol{z})] + D_{KL}(q(\boldsymbol{z}|\boldsymbol{x})||p(\boldsymbol{z})) + \mathcal{H}[p(y|\boldsymbol{z})] - E_{q(\boldsymbol{z}|\boldsymbol{x})}[\log p(y|\boldsymbol{z})]. \quad &\text{(10b)}
\end{aligned}$$

This entropy regularisation encourages the classification boundaries to be located in low sample density regions [30] in the latent space, which improves the generalisation performance of the model. As shown below (see Results section), the semi-supervised classification accuracy was very convincing, when using this entropy regularisation.

During training, mini-batches $b = 1, \ldots, B$ from the cell line $\mathcal{D}_{cl}^{(b)} \subset \mathcal{D}_{cl}^{Train}$ and tumour data $\mathcal{D}_{tu}^{(b)} \subset \mathcal{D}_{tu}^{Train}$ are used to minimise

$$\sum_{\boldsymbol{x}_c \in \mathcal{D}_{cl}^{(b)}} \mathcal{U}(\boldsymbol{x}_c) + \sum_{(\boldsymbol{x}_t, y_t) \in \mathcal{D}_{tu}^{(b)}} \mathcal{S}(\boldsymbol{x}_t, y_t) \tag{11}$$

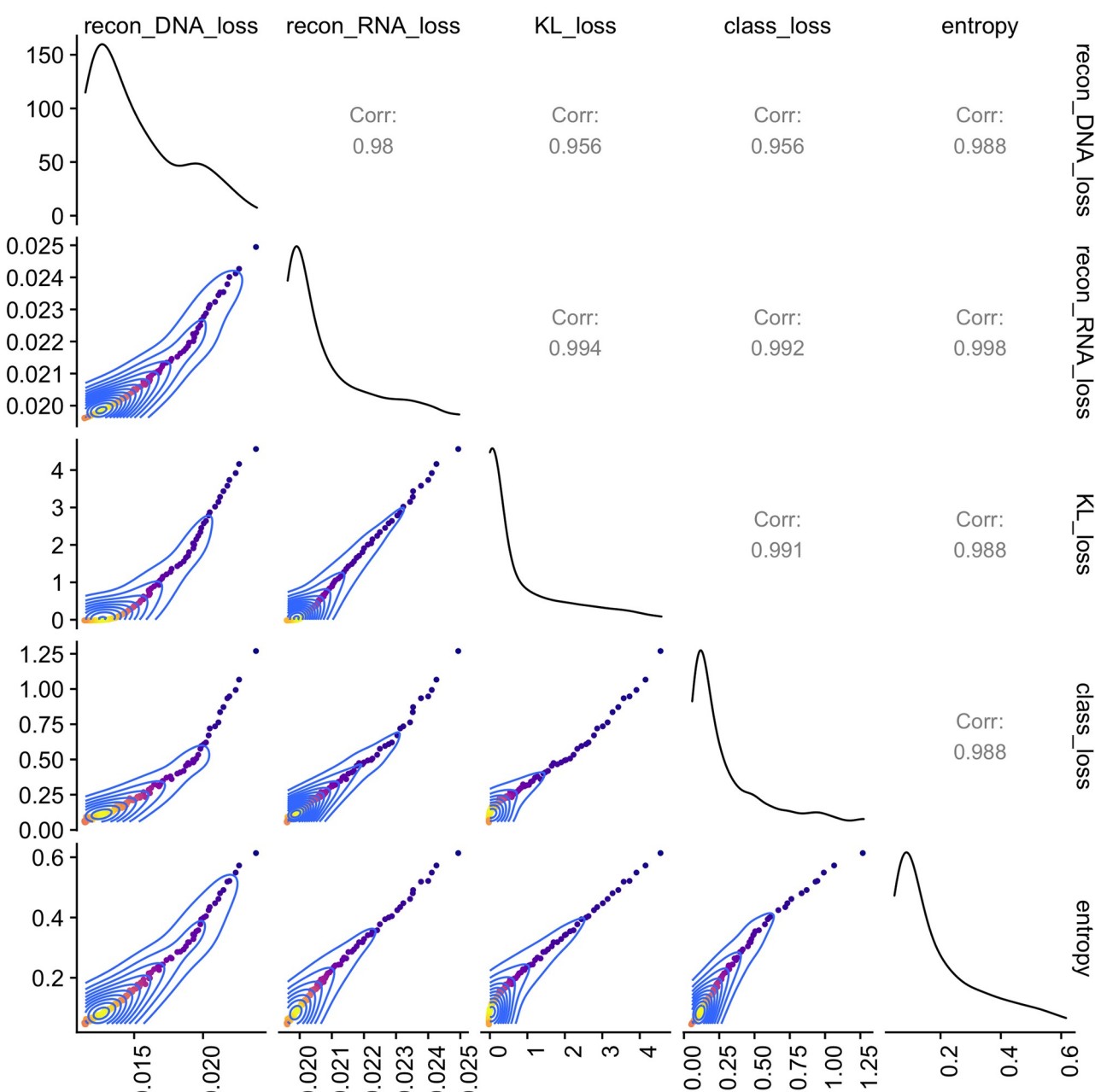

**Fig 2. Joint optimisation of the reconstruction loss, the KL divergence, entropy and the classification loss with the MFmap loss function.** The plot shows the pairwise correlation of different terms in the MFmap loss function Eq (10) during different training epochs.

over different epochs. To check whether all terms in the MFmap loss function in Eq (10) can be jointly optimised, we recorded the values of each term in each training epoch and calculated their pair-wise correlations. The reconstruction loss $-E_{q(z|x)}[p(x|z)]$, the KL-divergence $D_{KL}(q(y|x)||p(z))$, the entropy $\mathcal{H}[p(y|z)]$ and the classification loss $-E_{q(z|x)}[\log p(x|z)]$ are highly correlated (Fig 2), what suggests that they are optimised simultaneously.

## Visualisation of individual samples

The MFmap latent representation $z$ can be used to visualise and organise the associations of individual tumour samples and cell lines (Fig 1(C)). Inspired by the visualisation concept of Onco-GPS (OncoGenic Positioning System) [31], we used the tumour samples with known subtypes to generate a reference map for the cancer subtypes. In this reference map, the components $z_1, \ldots, z_h$ of the latent representation are presented as a graph with $h$ corner points in a plane. The location of these corner points is determined by multidimensional scaling and is chosen so as to reflect the distances in the $h$-dimensional latent space as good as possible (see S1 File for details). An individual tumour sample can now be visualised as a point located in the area between the corner points. The location of such a point is given by a superposition of the corner positions weighted by the latent representation magnitudes of individual samples. In addition, the subtypes of the tumour samples are colour coded. The contour lines and the background colour shading represent the sample density in the region.

Once the reference map is established, individual cell lines can be projected to this map, where the colour of each dot encodes the subtype *predicted* by the MFmap classifier. This projection is based on the latent representation values of the cell line samples. Since our aim is to analyse the fidelity of a cell line as an oncological model for a given tumour or a cancer subtype, we name our framework the model fidelity map (MFmap).

## Results

### Evaluating the MFmap classification and generative performance

A direct evaluation of the MFmap subtype prediction for cell lines is impossible because there are no ground truth labels available. However, the classification accuracy on an unseen test dataset of bulk tumours provides an indirect evaluation of the subtype prediction performance. In Table 2 we used 20% of the tumour samples as independent test set and evaluated the classification performance using four multi-class classification metrics: overall accuracy, weighted precision, weighted recall, and weighted $F_1$ score. Similar results can be obtained, when 10% of the tumour samples are used for testing (see Table 1 in the S2 File). We also tested the effect of increasing the latent space dimension $d$ and found that the classification accuracy was typically not higher, indicating that our choice of setting $d$ equal to the number of cancer subtypes did not impair the classification accuracy (see Table 2 in the S2 File).

The good classification results for GBMLGG are intriguing, because the G-CIMP-High, G-CIMP-Low and LGm6-GBM subtypes were derived from methylation data [32], which

**Table 2. MFmap subtype classification performance estimated for unseen tumour samples.** Here, 20% of the bulk tumour data were randomly selected as an independent test set.

| accuracy | precision | recall | $F_1$ score | organ |
|---|---|---|---|---|
| 0.97 | 0.97 | 0.97 | 0.97 | BRCA |
| 0.96 | 0.96 | 0.96 | 0.96 | COADREAD |
| 1.00 | 1.00 | 1.00 | 1.00 | ESCA |
| 0.99 | 0.99 | 0.99 | 0.99 | GBMLGG |
| 0.91 | 0.92 | 0.91 | 0.91 | HNSC |
| 0.96 | 0.96 | 0.96 | 0.96 | LUAD |
| 0.94 | 0.95 | 0.94 | 0.94 | LUSC |
| 0.97 | 0.97 | 0.97 | 0.97 | PAAD |
| 1.00 | 1.00 | 1.00 | 1.00 | SKCM |
| 0.96 | 0.96 | 0.96 | 0.96 | UCEC |

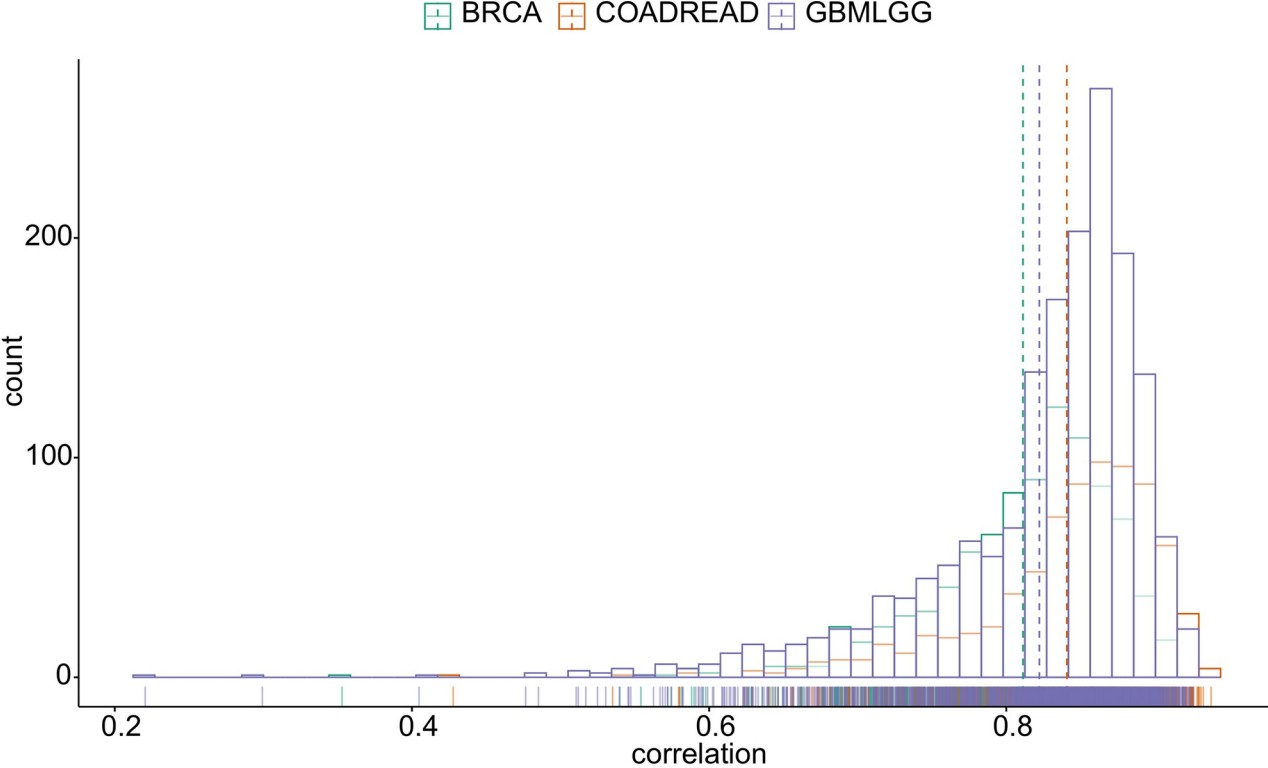

**Fig 3. The generative performance of MFmap.** The histogram shows sample-wise correlation coefficients between input features (DNA and RNA views) and reconstructed features output by the MFmap decoder.

were not used to train MFmap. This indicates that MFmap is able to extract DNA and RNA patterns reflecting features originally derived from different methylation status.

In addition, we tested how well the MFmap autoencoder part reconstructs the molecular features $x$. To this end, we first sampled a latent representations from the encoder $q(z|x)$ for a given input $x$ from the real data. Then, we correlated these original molecular features with the output sampled from the decoder distribution $p(x|z)$. The histogram of Pearson correlation coefficients in Fig 3 shows a high input-output correlation for most molecular features for three exemplary cancer types: breast invasive carcinoma (BRCA), colorectal adenocarcinoma (COADREAD) and glioblastoma multiforme and lower grade glioma (GBMLGG). Taken together, MFmap can combine very good classification accuracy with good generative performance.

Future applications of MFmap will include the analysis of query samples input to a reference model trained on a large data set. To check how well MFmap can perform in such a setting, we checked various measures for the quality of integrating these data from different sources [33–35]. Since this is not the focus of this paper, we have relegated the very promising results to the Supporting Information (see S2 File).

## Selecting the optimal cell line for a given tumour

The heatmaps in Fig 4 represent pairwise cell line by tumour dissimilarity matrices for three exemplary cancer types BRCA, COADREAD and GBMLGG. In addition, the subtypes of bulk tumours annotated from [32, 36, 37] and the subtypes of cell lines predicted by the MFmap

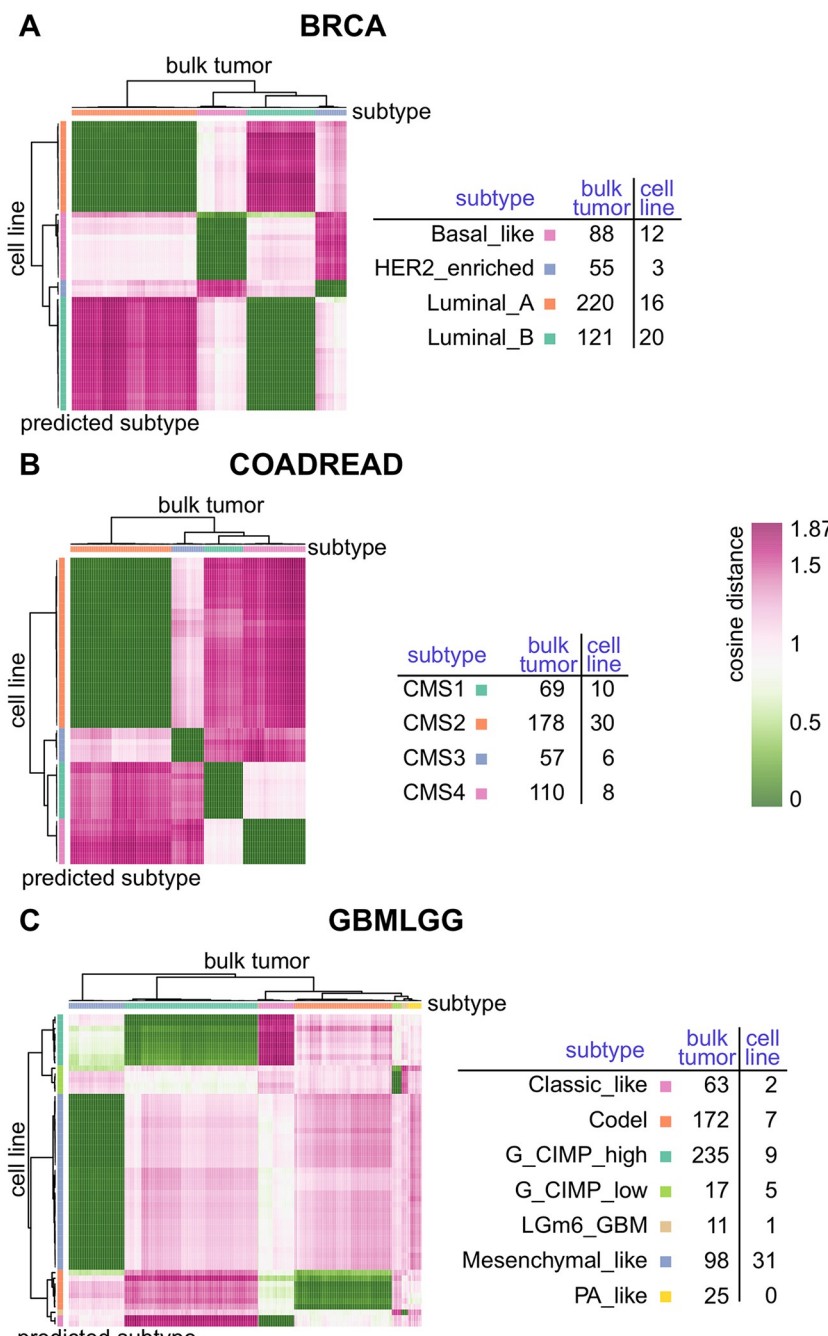

**Fig 4. Pairwise dissimilarity between CCLE cell lines and TCGA bulk tumours.** The colour coding in the heatmaps indicates the pairwise dissimilarity which was obtained from the latent representations of cell lines and tumours for the three exemplary cancer types (A) breast invasive carcinoma (BRCA), (B) colorectal adenocarcinoma (COADREAD) and (C) glioblastoma multiforme and lower grade glioma (GBMLGG). Tumours (columns) and cell lines (rows) were clustered according to the dissimilarity score, which ranges from 0 (very similar) to 2 (very dissimilar). The subtype classification of each cell line was predicted from the classification layer of the MFmap neural network. The tables display the sample size for the different subtypes or predicted subtypes.

classifier are displayed. For a better visualisation, cell lines and tumours are clustered based on their pairwise cosine dissimilarity scores. The similarity of a cell line $c$ to a tumour $t$ is defined as the cosine of the angle between their latent representations $z_c$ and $z_t$. Accordingly, the dissimilarity between $c$ and $t$ is defined as $d(c, t) = 1 - z_c \cdot z_t / \| z_c \| \| z_t \|$. A dissimilarity of $d(c, t) = 0$ indicates perfect alignment between the latent representations of the cell line and the tumour, whereas a dissimilarity $d(c, t) = 1$ indicates orthogonal latent representations. The highest dissimilarity of $d(c, t) = 2$ would be achieved for antipodal latent vectors. Based on this dissimilarity matrix, researchers can select the best cell lines for a given tumour or a given tumour subtype. And, vice versa, the relevance of promising experimental results observed *in vitro* can be checked by selecting a subset of tumours most likely resembling the cell line characteristics. The pairwise dissimilarity matrices between TCGA bulk tumours and CCLE cell lines and cell line subtype predictions for all tumour types listed in Table 1 are provided on our website (http://h2926513.stratoserver.net:3838/MFmap_shiny/).

These results also indicate, for which subtypes suitable cell line models exist and for which subtypes cell lines should be prioritised for future *in vitro* model development [21]. Each BRCA subtype is represented by at least three cell lines (Fig 4(A)) and the heatmap shows that these cell lines are very similar to the corresponding tumours of the same subtype. However, only three cell lines represents the HER2-enriched subtype. The four subtypes of COADREAD tumours are also well represented by at least six highly similar cell lines in CCLE (Fig 4(B)).

For GBMLGG, the Mesenchymal-like tumour subtype is represented by 31 cell lines with high similarity scores. Many TCGA tumour samples have the molecular subtype Codel and G-CIMP-high, but they are only represented by seven and nine cell lines, respectively. Only two cell lines were classified as Classic-like and a single cell line has the predicted subtype LGm6-GBM. The PA-like tumour subtype is not represented by any cell line.

## Predicting drug sensitivity in cancer patient sub-cohorts using MFmap and *in vitro* drug screens

Predicting patient therapeutic response is one important goal of subtype stratification. To explore the translational potential of the subtypes predicted by MFmap we estimated the association between predicted subtypes and drug sensitivity of all compounds available in the CTRP dataset [18]. For each cancer type listed in Table 1 and each compound, we compared the drug sensitivity among different cell line subtypes predicted by the MFmap classifier. Drug sensitivity is quantified in CTRP by the area under the dose response curve (AUC). We used an ANOVA to test for differences in the mean AUC among the predicted subtypes. At a false discovery rate (FDR) cutoff of 25%, we found 18, six and 16 compounds in BRCA, GBMLGG and UCEC to show significant subtype specificity, respectively. For the other seven cancer types in Table 1, there are no significant AUC differences across the different subtypes. Note that the sample size per subtype is very small, which might explain why statistically significant results can only be obtained for three cancer types.

For BRCA, the compound with the strongest association between subtype and drug sensitivity is Lapatinib (ANOVA p-value = 2.95e-05). Lapatinib is a tyrosine kinase inhibitor used in combination therapy for HER2-positive breast cancer [38]. Our results suggest that cell lines of molecular subtype HER2-enriched are more sensitive to Lapatinib treatment (Fig 5 (A)) in comparison to other three subtypes. Although there are only three cell lines representing the HER2-enriched subtype, this finding is in line with the known inhibitive mechanism of Lapatinib on the HER2/neu and epidermal growth factor receptor (EGFR) pathways. This result highlights the potential of MFmap as a tool for translating *in vitro* drug screening results to patient sub-cohorts. Our analysis also suggests that larger sample sizes and a better coverage

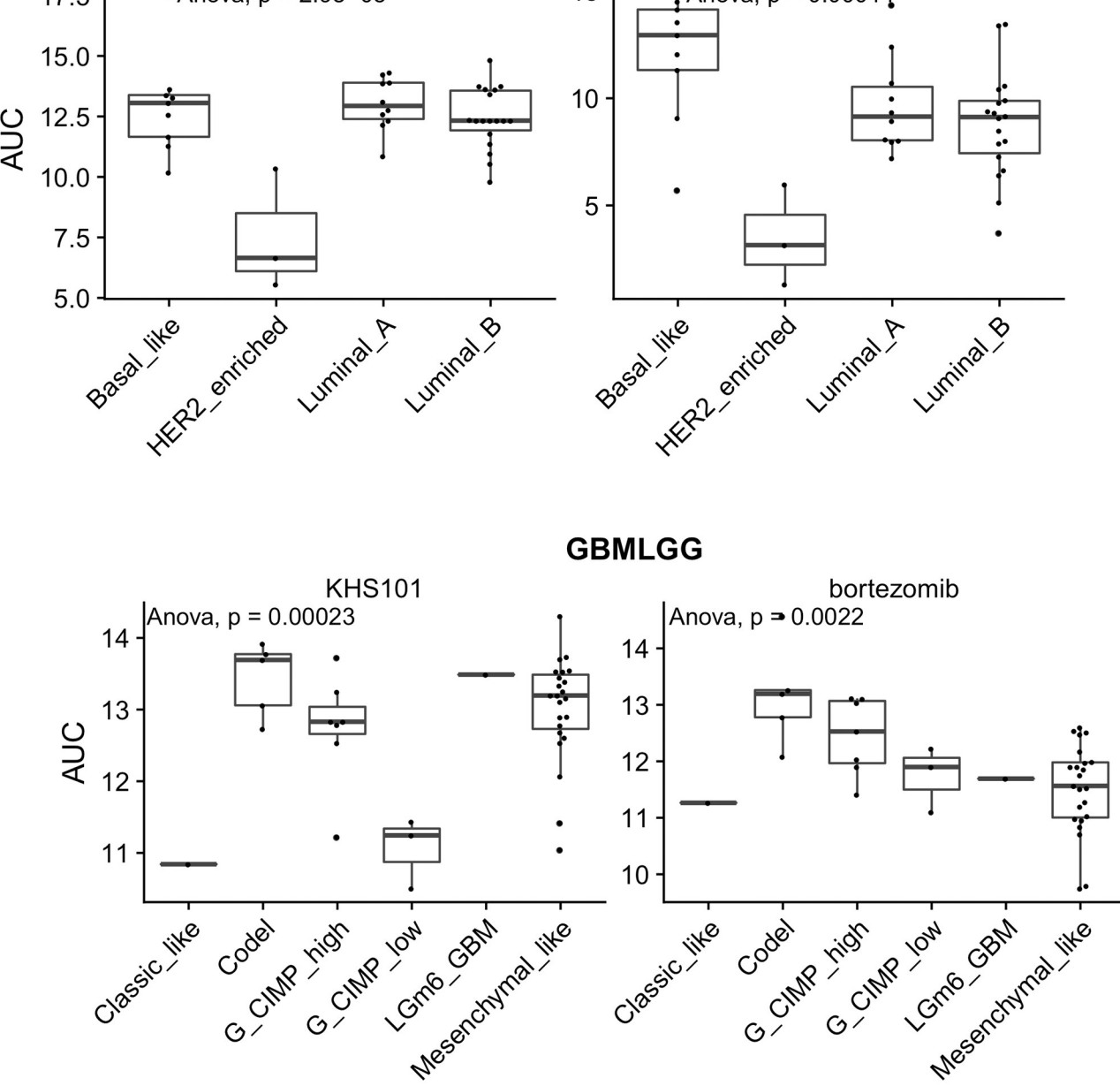

**Fig 5. Cancer subtype specific drug sensitivity of CCLE cell lines.** The subtypes of breast invasive carcinoma (BRCA) cell lines respond differentially to the compounds Lapatinib and Olygomycin A. Treatment response to the compounds KHS101 and Bortezomib in of glioblastoma multiforme and lower grade glioma (GBMLGG) cell lines is subtype specific. The drug sensitivity is summarised by the area under the dose response curve (AUC) and p-values refer to an ANOVA of the AUC differences among different subtypes.

of underrepresented subtypes are essential to increase the statistical power for detecting subtype specificity from cell line drug screens.

Another drug with significant variations of the AUC values across the different BRCA subtypes is Oligomycin A (ANOVA p-value = 1.39e-4), a compound targeting oxidative

phosphorylation via an inhibition of the ATP synthase. The potential of Oligomycin A as a therapeutic compound to prevent metastatic spread in breast cancer has recently been highlighted [39]. The results in Fig 5(B) suggest that treatment with Oligomycin A might be most efficient for the HER2-enriched and Luminal A or Luminal B subtypes.

The drug sensitivities of KHS101 and Bortezomib are significantly associated with GBMLGG subtypes (KHS101: ANOVA p-value = 2.3e-04; Bortezomib: ANOVA p-value = 2.3e-04). The synthetic small molecule KHS101 was shown to promote tumour cell death in diverse glioblastoma multiforme cell line models [40]. Our analysis suggests that the G-CIMP-low subtype is more sensitive to KSH101 treatment (Fig 5(C)) compared to the other six GBMLGG subtypes. G-CIMP-low is an IDH mutant glioma subtype with poor clinical outcome in recurrent glioma [32].

Bortezomib targets the ubiquitin-proteasome pathway and is used for the treatment of multiple myeloma, but has also been discussed as treatment for glioma [41]. Our results in Fig 5 (D) show that the Codel and G-CIMP-high subtypes have larger AUCs. The results for LGm6-GBM and Classic-like are not conclusive because there are not enough cell lines representing these subtypes.

## Biological characterisation of latent representations learnt by MFmap

The pattern of MFmap learnt latent representations $z$ can be used as a signature for cancer subtypes. For example, in BRCA, the basal-like subtype is characterised by a pattern of low values of components $z_1$ and $z_4$ and high values of $z_2$ and $z_3$ (Fig 6(A)). HER2-enriched tumours are characterised by high values of $z_1$ and $z_3$ and $z_4$. Luminal A and B subtypes can be distinguished by $z_4$. Similarly, cancer subtypes in COADREAD and GBMLGG are highly associated with their latent representations learnt by MFmap (Fig 6(B) and 6(C)).

To further investigate the biological meaning of the latent representations we analysed the association between $z$ and pathway activities in TCGA reference datasets. We used single sample gene set enrichment analysis (ssGSEA) [42] to assess sample-wise pathway activities. The pathway signatures were compiled from several sources including 10 curated oncogenic signalling pathways [43], 19 curated specific DNA damage repair (DDR) pathways [44], 14 expert-curated specific DDR processes and DDR associated processes [45]. This collection was combined with MsigDB (v7.0) [46] chemical and genetic perturbations (CGP) and canonical pathways (CP) collections (MsigDB C2 collection) and MsigDB (v7.0) hallmark gene sets (MsigDB H collection). The degree of associations was quantified by the information coefficient and the Pearson correlation coefficient and the statistical significance was assessed by permutation tests. To tackle class imbalance in the different subtypes, we applied SMOTE upsampling [47].

We used COADREAD as a proof of concept, because it has four well characterised molecular subtypes CMS1-CMS4 [37]. The CMS1 subtype is characterised by micro-satellite instability (MSI), whereas CMS4 tumours are micro-satellite stable. The CMS4 subtype is also distinguished from CMS1 by epithelial mesenchymal transformation (EMT) characteristics, accompanied by prominent stromal invasion and angiogenesis. These mutually exclusive characteristics are clearly reflected in the magnitudes of the latent representation components. The top gene sets associated with component $z_2$ are "WATANABE COLON CANCER MSI VS MSS UP" and "KOINUMA COLON CANCER MSI UP", whereas $z_4$ is associated with the activity of gene sets annotated as "HALLMARK ANGIOGENESIS" and "HALLMARK EPITHELIAL MESENCHYMAL TRANSITION". Clearly, high values of $z_2$ are a characteristics of the CMS1 subtype, whereas high values of $z_4$ are a distinctive feature of CMS4 tumours. This example illustrates that a meaningful way to guide biological interpretation of the latent representations is to associate them to single sample pathway activity.

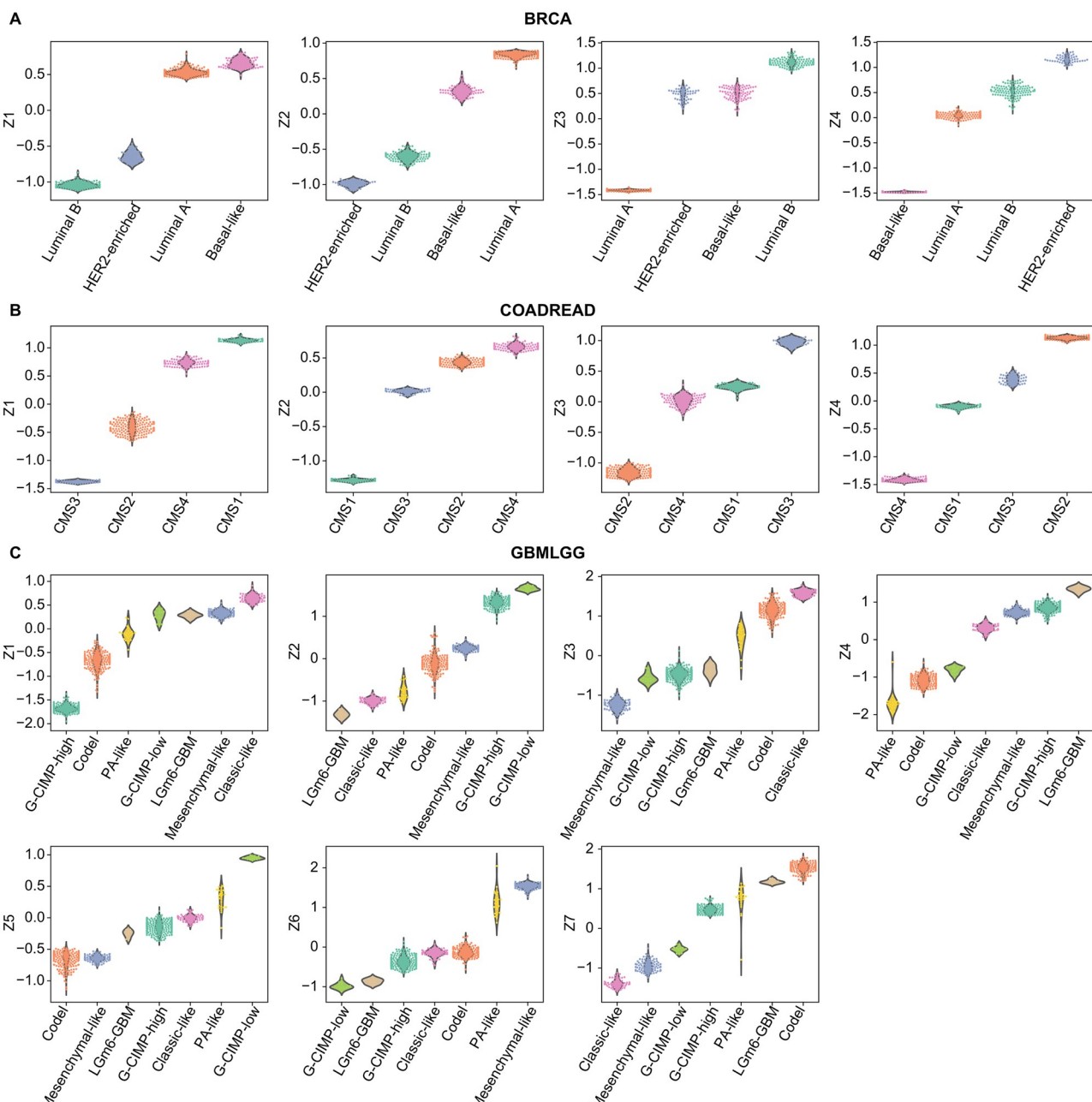

**Fig 6. Association of MFmap latent representations and cancer subtypes.** The dimension of the latent representation $h$ is set to the number of cancer subtypes. The boxplots display latent representations of different subtypes of TCGA samples in the three exemplary cancer types (A) breast invasive carcinoma (BRCA), (B) colorectal adenocarcinoma (COADREAD) and (C) glioblastoma multiforme and lower grade glioma (GBMLGG). Cancer subtypes are colour encoded and sorted by their median latent representations.

The same method was applied to annotate latent representations of GBMLGG (Fig 7(A)), which has seven subtypes [32]. The Mesenchymal-like and PA-like are stratified by gene expression profiles and the G-CIMP-high, G-CIMP-low and LGm6-GBM are methylation based. The Codel subtype describes IDH-mutant samples harbouring a co-deletion of chromosome arm 1p and 19q. Many pathways associated with latent representation $z_1$ are related to the neurotransmitter release cycle, which is also a characteristics of the Verhaak proneuronal

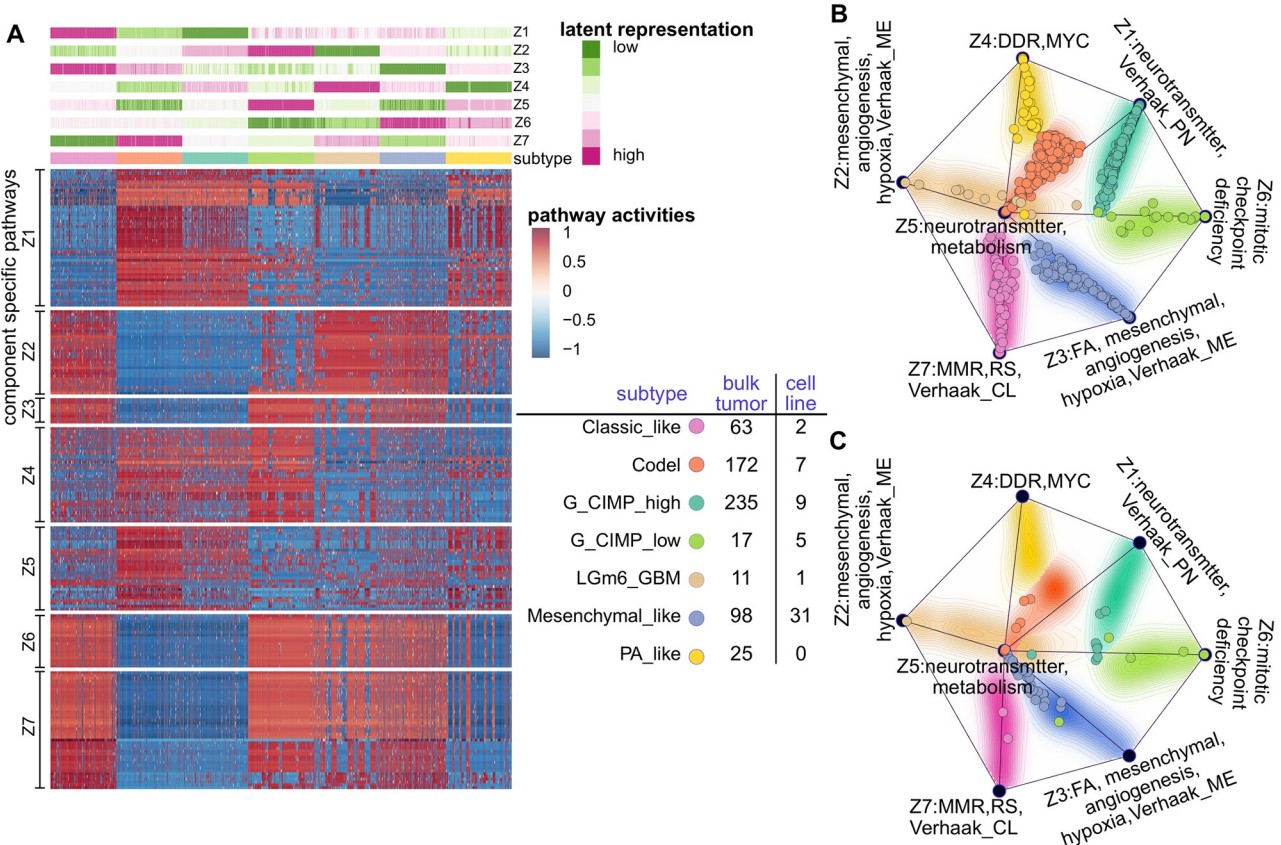

**Fig 7. Characterising the MFmap learnt latent representations in glioblastoma multiforme and lower grade glioma (GBMLGG).** (A) The top heatmap shows the latent representation $z$ of TCGA tumour samples (columns). The tumour samples are ordered based on a hierarchical clustering of $z$ and their subtypes are colour encoded. The heatmap at the bottom displays sample-wise pathway activities that are significantly associated with the latent representations $z_1, \ldots, z_7$. Pathway activities were computed using the ssGSEA algorithm [42]. For better visualisation, we upsampled the input data of MFmap and ssGSEA to get a balanced sample size in each subtype. (B) The MFmap reference map is formed by projecting the latent representations $z$ of bulk tumours into two dimensions using multidimensional scaling. It consists of seven dominant components represented by black nodes. The length of their connections is given by the Euclidean distance of the dominant components in the latent space. The annotation of the seven dominant nodes is based on the correlation between $z$ and pathway activity scores (see A). The background colour encodes sample subtypes, and the background contour encodes sample density. Individual bulk tumours are displayed as dots on the MFmap reference map. (C) Cell line samples are projected to the MFmap reference map. In both (B) and (C), the subtype of bulk tumours and predicted subtype of cell lines are colour coded. Subtype specific sample size for bulk tumours and cell lines is reported in the legend table.

subtype [48]. Pathways correlated to latent representation $z_2$ are related to the mesenchymal cell type, hypoxia and angiogenesis, which characterises the Verhaak mesenchymal subtype. The activity of the Fanconi Anemia (FA) DNA repair pathway is highly correlated with latent representation $z_3$. DNA damage response deficiency and amplified oncogenic MYC signalling characterises tumours with large values of latent representation $z_4$. Latent representation $z_5$ is related to the neurotransmitter release cycle and dysfunctional metabolism; latent representation $z_6$ to mitotic checkpoint deficiency. Many pathways associated with latent representation $z_7$ are involved in mismatch repair deficiency, replication stress and cell cycle disregulation and also related to the classical subtype in the earlier classification of Verhaak [48].

Individual samples and their relationships can be displayed in the MFmap reference map (Fig 7(B)), a visualisation tool adapted from OncoGPS [31]. Here, the seven corners of the map correspond to the respective latent representations $z_1, \ldots, z_7$ in GBMLGG. The corner locations are determined by multidimensional scaling on the latent representations of bulk

tumours. Individual bulk tumour samples are displayed as dots in the regions between the corner points with locations determined by a weighted vector sum of the seven corner locations (see S1 File for details). The subtype of each tumour sample is indicated by colours. The density of the tumour samples of a given subtype is depicted by the contour lines and the corresponding colour shading. Fig 7(B)) shows that samples of the same subtype clustered together and the inter-cluster distance is large. Projecting cell lines to the MFmap reference map (Fig 7(C)) helps to visualise the relationship between their predicted subtypes and their latent representations.

## Modelling cellular state transformations using latent space arithmetics

Cancerous neoplasms undergo various biochemical changes during cancer evolution and in response to selective pressure. One example is the transition from a proneural to a mesenchymal phenotype in glioblastoma, which is characterised by acquired therapeutic resistance and more aggressive potential [49]. In the DNA methylation based subtype classification of [32], the G-CIMP-high methylation phenotype tends to have the proneural molecular subtype [48] (see Fig 7(B)). Given that the latent representations learnt by MFmap clearly distinguish these different subtypes, we asked, whether the generative nature of the semi-supervised VAE can also be exploited to study such cancer subtype transformations.

To this end, we used the latent representations of the G-CIMP-high tumours and the Mesenchymal-like tumours (see Fig 7(B)) and computed the centroid vectors $\bar{z}_{\mathrm{G-CIMP-high}}$ and $\bar{z}_{\mathrm{Mesenchymal-like}}$ for the corresponding tumour samples. The difference $\delta = \bar{z}_{\mathrm{Mesenchymal-like}} - \bar{z}_{\mathrm{G-CIMP-high}}$ was used as a latent perturbation vector. By adding $\delta$ to the latent representation of each G-CIMP-high tumour (Fig 8(A)) we obtained the latent representation of *in silico* samples (Fig 8(B)), which are located in the "Mesenchymal-like region" of the reference map. We used these latent representation vectors of the *in silico* samples as input to the decoder of the MFmap network. We then checked, whether key molecular features of real Mesenchymal-like samples are reflected by these generated samples. Based on the available biological knowledge, we focussed on the most prominent onco-markers of the G-CIMP-high subtype: mutation status of the alpha thalassemia/mental retardation syndrome X-linked (ATRX), isocitrate dehydrogenase (IDH) and TP53 genes. The original G-CIMP-high tumours show a high propensity towards mutations in these genes, indicated by relatively higher network smoothed mutation scores (Fig 8(C)), although not all samples are necessarily harbouring these mutations. In contrast, the predicted mutation scores for the perturbed *in silico* samples in Fig 8(B) are much lower, indicating a lower propensity to IDH1, ATRX or TP53 mutations. This is in agreement with the observed tendency of Mesenchymal-like tumours for these mutations [49]. This example not only highlights the good generative performance of MFmap but also hints at potential applications on integrative analysis of cancer evolution dynamics.

## Discussion

Limited success in translating *in vitro* therapeutic markers to clinical applications highlights that not all cell lines are good models for a given cancer subtype. Selecting the most appropriate cell line for a given tumour or a set of tumours is crucial for understanding cancer biology and developing new anti-cancer treatments. Here, we provide a computational framework and a resource for cancer researchers to select the best cell lines for a TCGA tumour or a cancer subtype from ten different cancer types (http://h2926513.stratoserver.net:3838/MFmap_shiny/ ). The quantitative similarity score enables researchers to judge, whether a given tumour or a subtype of tumours is well represented by a cell line.

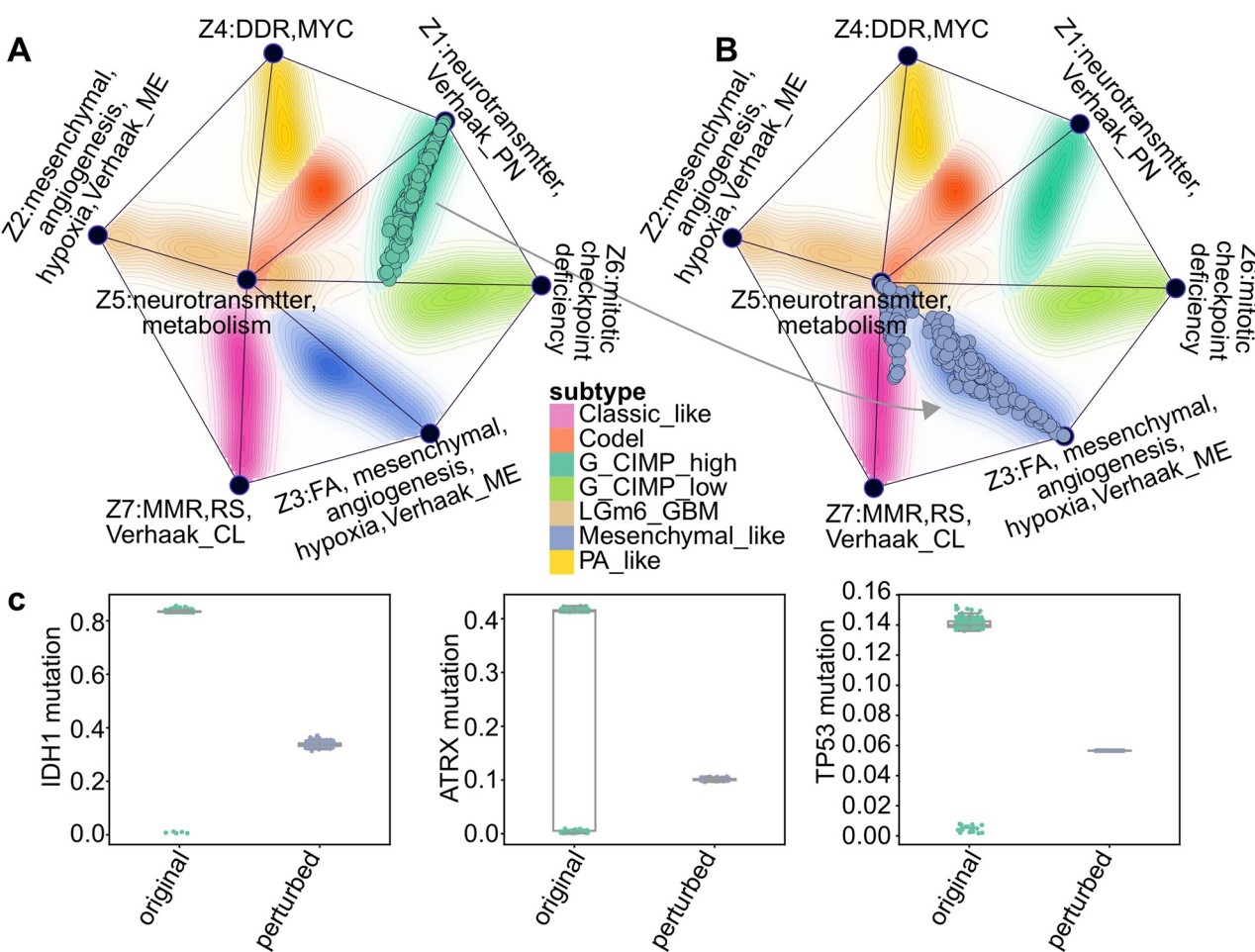

**Fig 8.** *In-silico* **perturbation analysis of cellular state changes during disease transformation from the G-CIMP-high to the Mesenchymal-like subtype in glioblastoma multiforme and lower grade glioma (GBMLGG).** (A) The G-CIMP-high tumours from TCGA are projected to the MFmap reference map. (B) By perturbing the latent representation vectors of these G-CIMP-high tumours we generate artificial tumour samples located in the Mesenchymal-like region of the MFmap reference map (compare Fig 7(B)). (C) Boxplots of the sample mutation status (network smoothed mutation scores) of marker genes IDH1, ATRX1 and TP53 before and after perturbation.

The assignment of cancer subtype labels to cell lines enables cell biologists to optimise experimental planning and to focus their research on clinically relevant model systems. We found that our semi-supervised MFmap model can classify tumours with a very high accuracy. Further analysis of drug sensitivity profiles supports that the subtype prediction for cell lines is biologically meaningful. Our analysis shows that HER2-enriched cell lines are most sensitive to Lapatinib, in agreement with prior knowledge about drug efficiency of this compound. As an example for the translation of *in vitro* pharmacogenomic data, we predict that the G-CIMP-low subtype is more sensitive to the new synthetic compound KHS101 compared to other GBMLGG subtypes.

Our finding that only BRCA, GBMLGG and UCEC show significant subtype specific drug sensitivity variation merits further investigation. One important reason is the small number of cell lines representing some cancer subtypes, which prevents us from finding statistically significant variations of drug sensitivity across the different subtypes. This highlights the need to prioritise cell line development for underrepresented disease variants [21]. However, it can not be ruled out that for some cancers the known subtype classifications are not predictive of drug

sensitivity. This suggests that clinically relevant subtype stratification should take into account drug sensitivity.

By embedding the original gene expression space, somatic mutation space and copy number space of bulk tumours and cell lines into a lower dimensional latent space, MFmap extracts latent features that are strongly associated with cancer subtypes. For COADREAD and GLMBGG, we have illustrated that the abstract latent representations can be annotated biologically using their associations with pathway activities. This makes the latent representations interpretable and allows to study the molecular and clinical heterogeneity of this disease. In principle, MFmap can be complemented by other modalities such as methylation or proteomics data. However, for our purpose we found that gene expression and DNA features in combination with the prior knowledge about tumour subtypes contains sufficient information.

Our proof of principle analysis of the transformation between two different tumour subtypes presents a new approach for studying tumour evolutionary processes in a more integrative way [50]. The small sample size of some multi-region sequencing or single-cell sequencing studies limits the ability to infer robust evolutionary patterns. By projecting these data to the MFmap reference map obtained from training on large sets of bulk tumour data one could deduce useful phenotypic information for individual patients. We believe that this can leverage information gathered in large cancer genomic studies like TCGA to guide personalised clinical decision making.

The MFmap is based on a new semi-supervised neural network architecture combining a basic VAE with an additional classifier. Such semi-supervised learning tasks are very common in the biomedical research field, because it is often easier to acquire a large number of measurements than to obtain the corresponding labels. Based on the good predictive and generative performance of MFmap together with the evidence provided here, that MFmap can learn biologically and clinically meaningful information, we are convinced that the MFmap model can be adapted to other semi-supervised tasks in oncology and beyond.

## Supporting information

**S1 File. Extended method details.**
(PDF)

**S2 File. Further evaluation of the MFmap performance.**
(PDF)

**S1 Data.**
(TXT)

## Author Contributions

**Conceptualization:** Xiaoxiao Zhang, Maik Kschischo.

**Data curation:** Xiaoxiao Zhang.

**Formal analysis:** Xiaoxiao Zhang, Maik Kschischo.

**Funding acquisition:** Maik Kschischo.

**Investigation:** Xiaoxiao Zhang, Maik Kschischo.

**Methodology:** Xiaoxiao Zhang, Maik Kschischo.

**Project administration:** Maik Kschischo.

**Resources:** Maik Kschischo.

**Software:** Xiaoxiao Zhang, Maik Kschischo.

**Supervision:** Maik Kschischo.

**Visualization:** Xiaoxiao Zhang.

**Writing – original draft:** Xiaoxiao Zhang, Maik Kschischo.

**Writing – review & editing:** Maik Kschischo.

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
