## [Decision Letter · Decision Letter 0]

13 Sep 2021

PONE-D-21-23744

MFmap: A semi-supervised generative model matching cell lines to tumours and cancer subtypes

PLOS ONE

Dear Dr. Kschischo,

Thank you for submitting your manuscript to PLOS ONE. After careful consideration, we feel that it has merit but does not fully meet PLOS ONE’s publication criteria as it currently stands. Therefore, we invite you to submit a revised version of the manuscript that addresses the points raised during the review process.

We look forward to receiving your revised manuscript.

Kind regards,

Tao Huang

Academic Editor

PLOS ONE

Journal Requirements:

“This work was supported by the FOR2800 research unit funded by the Deutsche Forschungsgemeinschaft (DFG project number 395736209).”

5. Please upload a new copy of Figure 4, 6 and 7 as the detail is not clear. Please follow the link for more information: https://blogs.plos.org/plos/2019/06/looking-good-tips-for-creating-your-plos-figures-graphics/

Reviewers' comments:

Reviewer's Responses to Questions

**Comments to the Author**

1. Is the manuscript technically sound, and do the data support the conclusions?

Reviewer #1: Yes

Reviewer #2: Yes

2. Has the statistical analysis been performed appropriately and rigorously? 

Reviewer #1: Yes

Reviewer #2: Yes

3. Have the authors made all data underlying the findings in their manuscript fully available?

Reviewer #1: Yes

Reviewer #2: Yes

4. Is the manuscript presented in an intelligible fashion and written in standard English?

Reviewer #1: Yes

Reviewer #2: Yes

5. Review Comments to the Author

Reviewer #1: The paper presents a variational autoencoder based model for simultaneous classification of cancer (subtypes) and representation learning. The model is applied to two publicly available cohorts (TCGA and CCLE). The methodology is presented convincingly and results appear sound. The authors have included links to 1) a Shiny server so that other can interactively query the results of the model and 2) to a Github repo with their source code so that others can apply the model to their data. This is very valuable.

One could perhaps get even more impact if a server running the code was set up and if datasets were provide so that other could benchmark their method against the proposed method. But this is nice to have not need to have.

So all in all this is good work. The paper can be recommended for publication more or less as is.

Reviewer #2: Zhang and Kschischo propose MFmap (model fidelity map) to simultaneously predict the cancer subtype of a cell line and its similarity to an individual tumour sample. A semi-supervised generative model, MFmap is a neural network architecture combining a basic VAE with an additional classifier. The use of the classifier to serves as a regulariser controlling the capability to learn a latent representation that is cancer subtype relevant is cleaver and does add to interpretability. However, the restriction of the dimension of the latent representation to the number of subtypes of cancer limits the algorithms application for biological discovery and potential denosing from unknown sources of technical variation. It would be interesting to know the effect of expanding to larger dimensionalizations; however, it reasonable to argue that is beyond the scope of the current work. Additionally, performance evaluations via “bake off” are not particularly informative, this work could benefit from the inclusion of a discussion of potential performance advantages over other methods. In particular, https://doi.org/10.1038/s41587-021-01001-7, https://doi.org/10.1016/j.cels.2019.04.004, and https://doi.org/10.1016/j.cels.2019.05.031 would be useful to place in the context of more widely used tools for integrated data analysis.

6. PLOS authors have the option to publish the peer review history of their article (what does this mean?). If published, this will include your full peer review and any attached files.

Reviewer #1: No

Reviewer #2: No

---

## [Author Response · Author response to Decision Letter 0]

25 Oct 2021

Response to the Editor

1. Please ensure that your manuscript meets PLOS ONE's style requirements, including those

for file naming...

We have used the PLOS Latex template and changed the naming of the files. 

2. Thank you for stating the following financial disclosure...

We have added "The funders had no role in study design, data collection and analysis, decision to

publish, or preparation of the manuscript." to the financial disclosure section. 

3. In your Data Availability statement...

We have prepared and uploaded a minimal data set. The relevant URLS of the TCGA data and the CCLE data are provided in the Supporting Information at the end of the main text. 

4. We note that you have included the phrase “data not shown” in your manuscript...

We provide the additional table as table S1 in the Supplemental file S1_file and refer to it in the main text. 

5. Please upload a new copy of Figure 4, 6 and 7 as the detail is not clear...

We have uploaded new copies of these figures and hope they are clear now. 

6. Please review your reference list to ensure that it is complete and correct...

We have checked the references. 

Response to the Reviewer #1:

Thank you for the very positive review. Following your suggestion, we we have added data to train MFmap to the Github repo to enable other researcher to compare their work to ours. 

Response to the Reviewer #2:

Thank you for the very positive review. 

We have added a file S1_file.pdf as Supplemental information and refer to it in the main text. There we provide the following additional evaluations of the MFmap:

 • In Table S1 we show the classification accuracy for a latent dimension of 100. You can see that the classification results are not really better, when we use a higher dimensional latent space. 

 • We provide additional performance evaluations for integrative analysis. We have focused on the scenario, that a reference model is trained with a large data set and a query data set (with a potential distribution shift) is fed into the reference model. We use the entropy of subtype mixing (ESM), the adjusted random index (ARI), the normalised mutual information (NMI) and the average silhouette width (ASW) mentioned in Lotfollahi et al. 2021 (https://www.nature.com/articles/s41587-021-01001-7) as performance measures for the consistency between reference and query. 

 • See also the sub-section “Evaluating the MFmap classification and generative performance” in the main text, were we refer to these additional performance evaluations in S1_file.pdf. 

 • We have cited the papers

 • https://doi.org/10.1038/s41587-021-01001-7

 • https://doi.org/10.1016/j.cels.2019.04.004

 • https://doi.org/10.1016/j.cels.2019.05.031

 suggested by you in the main text. 

suggested by you in the main text.

---

## [Decision Letter · Decision Letter 1]

25 Nov 2021

MFmap: A semi-supervised generative model matching cell lines to tumours and cancer subtypes

PONE-D-21-23744R1

Dear Dr. Kschischo,

We’re pleased to inform you that your manuscript has been judged scientifically suitable for publication and will be formally accepted for publication once it meets all outstanding technical requirements.

Kind regards,

Tao Huang

Academic Editor

PLOS ONE

Additional Editor Comments (optional):

Reviewers' comments:

Reviewer's Responses to Questions

**Comments to the Author**

1. If the authors have adequately addressed your comments raised in a previous round of review and you feel that this manuscript is now acceptable for publication, you may indicate that here to bypass the “Comments to the Author” section, enter your conflict of interest statement in the “Confidential to Editor” section, and submit your "Accept" recommendation.

Reviewer #3: All comments have been addressed

2. Is the manuscript technically sound, and do the data support the conclusions?

Reviewer #3: Yes

3. Has the statistical analysis been performed appropriately and rigorously? 

Reviewer #3: Yes

4. Have the authors made all data underlying the findings in their manuscript fully available?

Reviewer #3: Yes

5. Is the manuscript presented in an intelligible fashion and written in standard English?

Reviewer #3: Yes

6. Review Comments to the Author

Reviewer #3: The authors have addressed all questions raised by the reviewers. I recommend publication of the manuscript.

7. PLOS authors have the option to publish the peer review history of their article (what does this mean?). If published, this will include your full peer review and any attached files.

Reviewer #3: No

---

## [Editor Report · Acceptance letter]

6 Dec 2021

PONE-D-21-23744R1 

MFmap: A semi-supervised generative model matching cell lines to tumours and cancer subtypes 

Dear Dr. Kschischo:

I'm pleased to inform you that your manuscript has been deemed suitable for publication in PLOS ONE. Congratulations! Your manuscript is now with our production department. 

Kind regards, 

on behalf of

Dr. Tao Huang 

Academic Editor

PLOS ONE